# Lactic Acid Permeability of Aquaporin-9 Enables Cytoplasmic Lactate Accumulation via an Ion Trap

**DOI:** 10.3390/life12010120

**Published:** 2022-01-14

**Authors:** Katharina Geistlinger, Jana D. R. Schmidt, Eric Beitz

**Affiliations:** Department of Pharmaceutical and Medicinal Chemistry, Pharmaceutical Institute, Christian-Albrechts-University of Kiel, Gutenbergstr. 76, 24118 Kiel, Germany; kgeistlinger@pharmazie.uni-kiel.de (K.G.); jschmidt@pharmazie.uni-kiel.de (J.D.R.S.)

**Keywords:** aquaglyceroporin, monocarboxylate transporter, l-lactic acid, l-lactate, proton, gradient, permeability, transport, accumulation, ion trap

## Abstract

(1) Background: Human aquaporin-9 (AQP9) conducts several small uncharged metabolites, such as glycerol, urea, and lactic acid. Certain brain tumors were shown to upregulate AQP9 expression, and the putative increase in lactic acid permeability was assigned to severity. (2) Methods: We expressed AQP9 and human monocarboxylate transporter 1 (MCT1) in yeast to determine the uptake rates and accumulation of radiolabeled l-lactate/l-lactic acid in different external pH conditions. (3) Results: The AQP9-mediated uptake of l-lactic acid was slow compared to MCT1 at neutral and slightly acidic pH, due to low concentrations of the neutral substrate species. At a pH corresponding to the pK_a_ of l-lactic acid, uptake via AQP9 was faster than via MCT1. Substrate accumulation was fundamentally different between AQP9 and MCT1. With MCT1, an equilibrium was reached, at which the intracellular and extracellular l-lactate/H^+^ concentrations were balanced. Uptake via AQP9 was linear, theoretically yielding orders of magnitude of higher substrate accumulation than MCT1. (4) Conclusions: The selectivity of AQP9 for neutral l-lactic acid establishes an ion trap for l-lactate after dissociation. This may be physiologically relevant if the transmembrane proton gradient is steep, and AQP9 acts as the sole uptake path on at least one side of a polarized cell.

## 1. Introduction

Channels of the aquaporin protein family (AQP) typically facilitate the transmembrane passage of water and small, neutral solutes driven by osmotic and chemical gradients. In humans, a plenitude of physiological functions are connected to cellular water permeability, including urine concentration, lung moistening, tear and saliva production, cerebrospinal fluid regulation, and even immune and tumor cell motility [1,2,3,4]. With regards to solute permeability, glycerol is probably the major AQP substrate, enabling shuttling between adipose tissue during lipolysis and the liver for gluconeogenesis [5,6]. Further, urea, ammonia, carbon dioxide, and hydrogen peroxide were shown to pass AQPs contributing to nitrogen and amino acid metabolism, pH regulation, and cellular signaling [7,8]. While the latter group of solutes may also take alternative routes across the cell membrane [9,10,11,12,13], AQPs appear unrivaled in their water and glycerol permeability functions.

One human aquaglyceroporin in particular, AQP9, was shown to additionally conduct l-lactic acid, i.e., the neutral, protonated form of the l-lactate anion, at considerable rates [14,15]. Despite the presence of dedicated l-lactate/H^+^ co-transporters of the human monocarboxylate transporter family MCT [16,17], indirect evidence is accumulating that the l-lactic acid permeability of AQP9 may play a role in the astrocyte-to-neuron l-lactate shuttle [18], and AQP9 expression levels may determine the severity of brain tumors [19]; for instance, during tumorigenesis, glioma cells increase AQP9 expression, and it has been suggested that the upregulation of AQP9 enhances the uptake of l-lactate as a nutrient for glioma cells, promoting tumor cell growth [20].

In order to judge the relative contribution of the l-lactic acid permeability of AQP9 vs. l-lactate/H^+^ transport via MCTs, one needs to consider the l-lactate protonation equilibrium in the given pH situation (Figure 1) [21,22]. The pK_a_ of l-lactic acid is 3.8, meaning that at a physiological pH of 7.4, one out of 4000 l-lactate molecules is present in its protonated, neutral l-lactic acid form, and can, thus, act as an AQP9 substrate. The proportion of neutral l-lactic acid increases with acidity, i.e., 1/400 at pH 6.4 and 1/40 at pH 5.4. MCTs, in turn, accept the l-lactate anion as a substrate and concomitantly use the transmembrane proton gradient as a driving force for l-lactate/H^+^ transport. The proportion of l-lactate anions is largely independent from the pH in the physiological range, i.e., 99.97% at pH 7.4, 99.75% at pH 6.4, and 97.5% at pH 5.4. We have previously found that AQP9 also attracts l-lactate anions to some degree, due to a cluster of positively charged arginine residues at the extracellular protein surface [15]. This enhances the probability for protonation and passage, as l-lactic acid corresponds to a shift in pH by about 0.5 units.

Here, we show that AQP9-facilitated permeation of neutral l-lactic acid and co-transport of l-lactate/H^+^ via MCT1 result in fundamentally different outcomes regarding the achievable substrate concentrations in a cell. While transport via MCT is equilibrative with respect to the extracellular l-lactate and proton concentrations, the l-lactic acid permeability of AQP9 enables an ion trap to be formed, leading to a large accumulation of intracellular substrate over the extracellular concentration in suitable transmembrane pH conditions.

## 2. Materials and Methods

### 2.1. Plasmids and Cloning

Expression constructs of human AQP9 (NCBI Gene ID 366) and human MCT1 (GenBank NM_001166496) in the modified pDR196 vector, containing the URA3 gene, were generated previously [15,23]. All constructs carried an N-terminal hemagglutinin tag and a C-terminal His_10_-tag.

### 2.2. Yeast Transformation and Culture

Yeast strain *Saccharomyces cerevisiae* W303-1A jen1Δ ady2Δ (MATa, can1-100, ade2-loc, his3-11-15, leu2-3,-112, trp1-1-1, ura3-1, jen1::kanMX4, ady2::hphMX4) without the endogenous monocarboxylate transport proteins Jen1 and Ady2, was kindly provided by M. Casal [24]. Yeast cells were transformed with the expression constructs or the empty plasmid using the lithium acetate method [25]. Transformed yeast cells were grown at 29 °C and 220 rpm, shaking in a synthetic defined (SD) medium containing 2% (*w/v*) glucose, plus l-histidine (20 mg L^−1^ final concentration), l-tryptophan (20 mg L^−1^), adenine (25 mg L^−1^) and l-leucine (100 mg L^−1^), but were lacking uracil for plasmid selection.

### 2.3. Radiolabel Transport Assays

The uptake assays were conducted as previously described [26]. Briefly, transformed cells were grown to an OD_600_ of 1 ± 0.1, harvested by centrifugation (4000× *g,* 5 min, 4 °C) and washed with 50 mL of ice-cold water (4000× *g*, 5 min, 4 °C). The cells were kept on ice for 30 min and then adjusted to an OD_600_ of 50 ± 5 in the assay buffer. The buffer contained 50 mM HEPES/50 mM Tris (pH 6.8), 50 mM MES/50 mM Tris (pH 5.8), or 50 mM citric acid/50 mM Tris (pH 4.8 and 3.8). Subsequently, 80 µL of yeast suspension, corresponding to 5.6 mg yeast cells, was prewarmed to 19 ± 1 °C for 2 min in a 1.5 mL microcentrifuge tube and mixed with 20 µL of substrate solution containing 5 mM sodium l-lactate or glycerol (1 mM final concentration) spiked with 0.002 µCi µL^−1^ of ^14^C-radiolabeled substrate (55 mCi/mmol). At the indicated time points, uptake was stopped by the addition of 1 mL of ice-cold water, rapid filtration through a GF/C glass microfiber filter (GE Healthcare, Solingen, Germany), and washing with 7 mL of ice-cold water. The filters were transferred to a scintillation vial containing 3 mL scintillation cocktail (Rotiszint^®^ eco plus, Roth), and incubated overnight. Each sample was counted for two minutes (Packard TriCarb 2900TR, Perkin Elmer, Rodgau, Germany) and the amount of cytosolic l-lactate or glycerol substrate was calculated from the counts per sample relative to the total substrate counted in buffer. Non-expressing yeast cells were assayed in the same conditions for background subtraction. Background diffusion was 0.133 ± 0.011 nmol mg^−1^ with glycerol, and 0.041 ± 0.020 nmol mg^−1^ (pH 6.8) or 0.135 ± 0.042 nmol mg^−1^ (pH 3.8) with l-lactate. For l-lactate efflux, yeast cells (600 µL suspension in total) were loaded using a 1 mM l-lactate inward gradient (pH 3.8, 8 min for MCT1, 3 min for AQP9 to achieve similar levels), and the amount of cytosolic l-lactate was determined from a 100 µL sample. Cells were collected (12,225× *g*, 10 s, 19 °C) and resuspended in 500 µL assay buffer of pH 6.8 or 3.8 without l-lactate. At the indicated time points, 100 µL samples were taken and analyzed.

### 2.4. Statistical Analysis

Data were plotted using SigmaPlot 14.5. Each experiment was conducted in biological triplicates (uptake) or duplicates (efflux) and technical duplicates. Data represent mean ± S.E.M.

## 3. Results

We expressed human AQP9 and human MCT1 in a well-established *Saccharomyces cerevisiae* yeast system that lacks endogenous **l-**lactate transporters (Δjen1 Δady2), which is used to conduct substrate uptake assays with minimal background (Figure 2a and Figure 3a indicate the low levels of transmembrane diffusion for glycerol and l-lactic acid) [15,26,27,28]. Western blots documenting the expression of the constructs are separately depicted in [15] for AQP9 and in [27] for MCT1, and by a joint plot in [22]. We used glycerol and l-lactate/lactic acid as substrates carrying a ^14^C-radiolabel. Since yeast is an ethanol-fermenting organism, it lacks l-lactate-producing enzymes, and the cytosolic l-lactate concentration is virtually zero [29]. Yeast cells can produce glycerol on demand as an osmolyte to counteract hypertonic media conditions [30,31]. Hence, we chose low-osmolyte culture and assay conditions to prevent cytosolic glycerol production. Therefore, we can assume zero-trans conditions for both glycerol and l-lactate, with the cytosol representing the trans compartment. To initiate uptake, we added 1 mM of substrate to the extracellular cis compartment in different buffer pH conditions, and subsequently determined the amount of radiolabel inside the cells over time by liquid scintillation counting. While measurements of l-lactate transport, when using typical expression systems, such as human cell lines or *Xenopus laevis* oocytes, are restricted to the seconds time scale due to rapid substrate metabolism [23], the yeast allowed us to monitor uptake up to 18 min, allowing us to make good estimates of the transport capacity [22,23]. Longer periods were prevented by decreasing the cell fitness under the assay conditions.

### 3.1. AQP9-Mediated Uptake of Glycerol Is Equilibrative and pH-Independent

AQP9, as a channel protein for neutral substrates, enables passage in both directions at the same velocity (k_1_ = k_−1_) [32]. The direction of the entropy-driven diffusion process is determined by the transmembrane gradient. An equilibrium is reached when the substrate concentration is equal on both sides of the membrane, shown in the following equation:k_1_ × *glycerol*_inside_ = k_–1_ × *glycerol*_outside_


At this stage, substrate molecules still pass the channel, yet with equal velocity in both directions, leading to a plateau in the uptake curve. The described behaviour is illustrated in Figure 2 for an external pH of 6.8, which matches the cytosolic pH of yeast cells [33], and for an acidic pH of 3.8 that generated a 1000× inward proton gradient. Yeast cells maintain the cytosolic pH close to neutral, even in acidic media via buffering [34], and a plasma membrane proton-ATPase [33]. We added 1 mM of ^14^C-labeled glycerol to yeast expressing human AQP9 and determined its uptake into the cells over time.

At early time points, glycerol uptake was rapid due to the initially maximal gradient, and slowed when approaching the equilibrium state. Both uptake curves appear to be highly similar because glycerol is a permanently neutral substrate, irrespective of the buffer pH. The levels of the plateaus were derived from single exponential rise-to-maximum fittings yielding 0.53 ± 0.01 nmol mg^−1^ yeast (pH 6.8) and 0.38 ± 0.02 nmol mg^−1^ yeast (pH 3.8). The observed difference of 27% is probably due to slight pH-dependent adaptations of the cell and vacuolar volume, particularly in harsh acidic conditions [35]. The glycerol load of the cells in the equilibrium, obtained in nmol mg^−1^ from scintillation counting, therefore translates into a cytosolic concentration of 1 mM, i.e., equal to the external buffer concentration.

### 3.2. MCT1-Mediated Uptake of l-Lactate Is pH-Dependent

The proton-driven, alternating access-type l-lactate transporter MCT1 binds its substrate in the anionic form, plus a proton [36]. The joint binding of l-lactate/H^+^ elicits a conformational change that triggers the transport protein to open at the trans side and release the substrate and its proton [37,38]; the kinetics can differ in the two transport directions (k_1_ ≠ k_−1_). Therefore, transport via MCT1 depends on two gradients across the membrane (l-lactate and H^+^), and the equilibrium state is reached when the proportions are balanced, as shown below:k_1_ × *lac^−^*_inside_ × H^+^_inside_ = k_−1_ × *lac^−^*_outside_ × H^+^_outside_

The dependency of MCT1-mediated transport of l-lactate from the transmembrane proton gradient is shown in Figure 3. At an external pH of 6.8, i.e., in the absence of a proton gradient across the yeast cell membrane, the transmembrane l-lactate gradient acts as the sole driving force. The proportion of the l-lactate anion at pH 6.8 is 0.999 mM vs. 0.001 mM of protonated l-lactic acid. Accordingly, at the transport equilibrium, the intracellular l-lactate concentration amounted to about 1 mM (the gray area in Figure 3a), equalling that of the external buffer.

Lowering the external pH to 5.8 (Figure 3b) and 4.8 (Figure 3c) established inward-directed proton gradients that drive l-lactate at increasing velocity and to higher cytosolic concentrations than present in the external buffer. In this pH range, the proportion of the l-lactate anion in relation to the neutral l-lactic acid is minimally affected (0.99 mM at pH 5.8 and 0.9 mM at pH 4.8; Figure 1). However, at the most acidic external pH of 3.8 (Figure 3d), the transport rate was markedly decreased. A pH of 3.8 corresponds to the pK_a_ of l-lactate. As a consequence, the concentration ratio of the l-lactate anion species and the neutral l-lactic acid form is 50% each (Figure 1). We extrapolated the plateau level to 1.7 ± 0.3 nmol mg^−1^, which is within the margin of error for the plateau at pH 4.8 (2.0 ± 0.1 nmol mg^−1^). Apparently, the 10-times steeper proton gradient compensated for the loss of 50% of the available l-lactate substrate. This is in line with the proton gradient effects at pH 5.8 and 4.8, at which a 10-fold higher proton gradient led to doubling of the cytosolic lactate concentration in the equilibrium.

### 3.3. AQP9-Mediated Uptake of l-Lactic Acid Causes Large Accumulation of l-Lactate Anions

Next, we assayed the AQP9-mediated uptake of l-lactic acid at an external total l-lactate/l-lactic acid concentration of 1 mM and in different pH conditions (Figure 4). At pH 6.8, uptake was hardly detectable over the assay time of 16 min (Figure 4a). The low availability of the neutral l-lactic acid substrate of 0.001 mM and the respective flat inward gradient under these conditions explains this behavior. With increasing acidity of the external buffer, however, more molecules of the l-lactic acid species were generated by protonation, amounting to 0.5 mM at pH 3.8 (see Figure 1). Accordingly, AQP9-facilitated l-lactic acid uptake became more prominent in more acidic external buffers (Figure 4b–d).

In the neutral and slightly acidic pH range, MCT1 exhibited higher transport velocities than AQP9 (Figure 5a), due to the acceptance of the predominant l-lactate anion (Figure 1) as a substrate [22]. AQP9-mediated l-lactic acid permeability became faster than MCT1 transport, however, when the external pH was adjusted to 3.8 (Figure 5a), i.e., at the reversal point of the l-lactate vs. l-lactic acid proportion (Figure 1).

Contrary to saturable MCT1-mediated transport (Figure 3), uptake via AQP9 appeared linear and did not reach an equilibrium state within the assay time (Figure 4). This shows that the achievable total concentration of cytosolic lactate was, by far, higher with AQP9 than with MCT1. The permeation of strictly neutral l-lactic acid (*LacH*) via AQP9 follows the same principle as glycerol, shown below:k_1_ × *LacH*_inside_ = k_−1_ × *LacH*_outside_

However, the availability of neutral l-lactic acid is coupled with the pH conditions via the protonation equilibrium, shown below: $${\mathrm{Lac}}^{–}\;+\;H^{+}\;⇄\mathrm{LacH}$$

This means that the ratio of the proton concentrations at both sides of a membrane, i.e., a transmembrane pH gradient, directly and fully translates into the accumulation factor of the protonatable substrate. Specifically, a transmembrane pH gradient of one log_10_ unit will lead to a ten-fold higher substrate concentration in the less acidic compartment, due to dissociation into the impermeable l-lactate anion (Figure 5b). At the lowest external pH in our assays of 3.8, i.e., 3 log_10_ units below the cytosolic pH in yeast (Figure 4d), we could expect a 1000-fold accumulation of cytosolic substrate in the form of impermeable l-lactate (Figure 5b). At pH 4.8, the accumulation factor was calculated to 100 with AQP9, while the l-lactate/H^+^ co-transporting MCT1 yielded its maximal accumulation factor of about four at this pH gradient (Figure 5b).

### 3.4. AQP9-Mediated l-Lactic Acid Permeability Establishes an Ion Trap

Eventually, we tested if the substrate selectivity of AQP9 for neutral l-lactic acid retains l-lactate in the cytosol, even when the cells are exposed to a substrate-free buffer, i.e., if a true ion trap is established. Therefore, we loaded AQP9- and MCT1-expressing yeast cells in ^14^C-labeled l-lactate/l-lactic acid buffer (1 mM total, pH 3.8) with adjusted loading times to reach cytosolic substrate levels of about 1 mM (AQP9: 3 min; MCT1: 8 min; see Figure 3d and Figure 4d). The cells were collected by centrifugation, and the supernatant was replaced by a pH 6.8 buffer without l-lactate/l-lactic acid. The amount of cytosolic ^14^C-l-lactate/l-lactic acid was probed over time by scintillation counting.

MCT1-expressing cells exhibited rapid efflux of the radiolabel (Figure 6a). At a cytosolic pH of 6.8 in the yeast, 99.9% of the substrate is present as the l-lactate species, resulting in an efflux curve (R^2^ = 0.757) with a similar shape and time scale to the l-lactate uptake assay (Figure 4), due to an equally steep, yet inverse, substrate gradient. In contrast, cells expressing AQP9 retained l-lactate in the cytosol at 1 ± 0.3 (R^2^ = 0.757), despite the established outward l-lactate gradient (Figure 6b). The effect was equal, even at an acidic external pH of 3.8, further illustrating the stability of the cytosolic pH (Figure 6b, inset). In order to pass the AQP9 channel, l-lactate needs to become protonated to form neutral l-lactic acid. However, at a cytosolic pH of 6.8, only a fraction of 0.1% (0.001 mM) of l-lactic acid is present. This experiment, therefore, directly depicts the ion trapping capability of AQP9, leading to the accumulation of charged l-lactate species. 

## 4. Discussion

The transmembrane facilitation of l-lactate/l-lactic acid by aquaglyceroporins and monocarboxylate transporters differs markedly in the following two basic characteristics: (i) In near-neutral pH gradient conditions, MCT1 enables high transport rates, whereas permeability via AQP9 is low. This is due to the usage of different protonation species of the substrate [22]. The MCT electrostatically attracts and transports the predominant l-lactate anion [36], while the AQP facilitates diffusion of the neutral l-lactic acid form [15], which is strongly underrepresented in the neutral pH range (Figure 1); (ii) MCT1-mediated substrate uptake is equilibrative with respect to the l-lactate and proton concentrations inside and outside of the cell [23], and reaches a maximum in the same order of magnitude as the external l-lactate concentration (Figure 7a). In contrast, the l-lactic acid permeability of AQP9 establishes a cytosolic ion trap for l-lactate that potentially leads to accumulation by several orders of magnitude, depending on the transmembrane pH gradient (Figure 7b).

A prerequisite for trapping weak acid anions is a stable, close-to-neutral cytosolic pH, which is also maintained at an acidic pH of the extracellular fluid. Since pH homeostasis is key for cell survival, the intracellular pH is controlled by the considerable chemical buffer capacity of the cytosol [34], and primary active transporters that actively expel excess protons into vacuoles (vacuolar H^+^-ATPases) [39] or into the extracellular space (plasma membrane H^+^-ATPases) [33]. These mechanisms counteract the uptake of protons from the importlof l-lactate/H^+^ or l-lactic acid.

Currently, it is unclear if physiological or pathophysiological situations exist in humans, in which cell growth or survival is linked to AQP9-mediated l-lactic acid permeability, and other possible AQP-related functions in tumors are discussed. Such roles might be involved in sustaining the energy metabolism by enabling the clearance of glycerol from tumor cells [22], or in AQP-driven tumor metastasis [40]. The case is more solid for microorganisms, specifically l-lactic acid-fermenting *Lactobacillus* bacteria that rely on efficient l-lactic acid export and express AQP9 homologues with high l-lactic acid permeability, even in close-to-neutral pH conditions [41,42]. In humans, circulating l-lactic acid is extensively used as a fuel, and was reported to be equally as relevant as glucose [43]. Particularly in the brain and in certain tumors, vital synergisms were identified between l-lactate-producing and -consuming cells [44]. In contrast to MCT-type transporters, the expression levels of AQP9 in the brain are typically low under physiological conditions [45]. However, upregulation of AQP9 was shown for glioma cells, and the expression levels were correlated to tumor severity [19,20]. In which scenarios may AQP9-mediated l-lactic acid permeability be of relevance for cell growth?

From our data, we can conclude that for a cell that expresses AQP9 as the sole pathway for l-lactic acid, a rather steep transmembrane pH gradient would probably be required in order to gain transmembrane facilitation rates that match the metabolic turnover of l-lactate in the cell. Such pH gradients may build up in confined spaces of close cell–cell interactions (Figure 7c), yet their presence remains to be shown. The simultaneous expression of AQP9 and a l-lactate-transporting MCT in a cell would prevent the formation of an ion trap because l-lactate would serve as an MCT transport substrate, prohibiting accumulation. However, if AQP9 and the MCT are localized at opposite apical/basolateral membranes of the cell, the l-lactic acid permeability of AQP9 would determine the directionality of l-lactate/l-lactic acid facilitation across the cell (Figure 7d). AQP9 would enable uptake, but prevent release, leaving only the MCT as the exit pathway at the opposite side of the cell. However, such cellular situations have not been shown yet by morphological or physiological studies, but are required before a role can be attributed to AQP9-mediated l-lactic acid permeability in cellular energy metabolism.

## Figures and Tables

**Figure 1 life-12-00120-f001:**
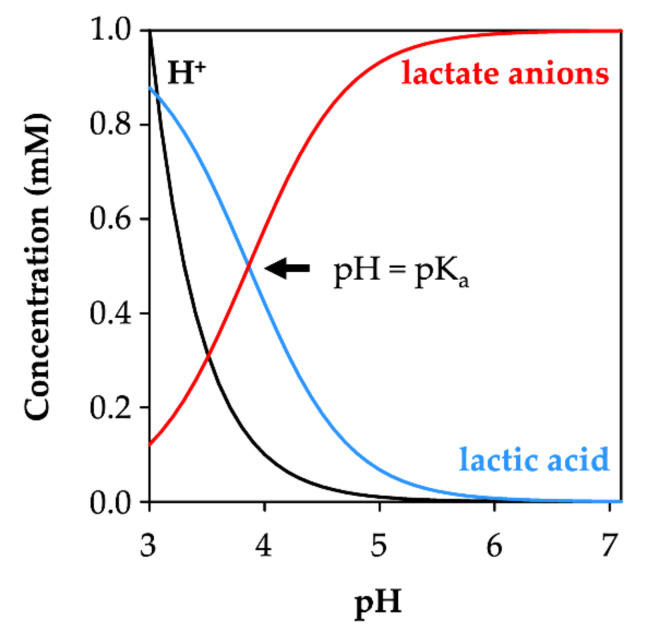
Concentration profiles of protons, lactate anions, and lactic acid over a pH range of 3–7 at a total lactate/lactic acid concentration of 1 mM. At pH = pK_a_ (3.8), lactate and lactic acid concentrations are equal (intersection of the red and blue curves).

**Figure 2 life-12-00120-f002:**
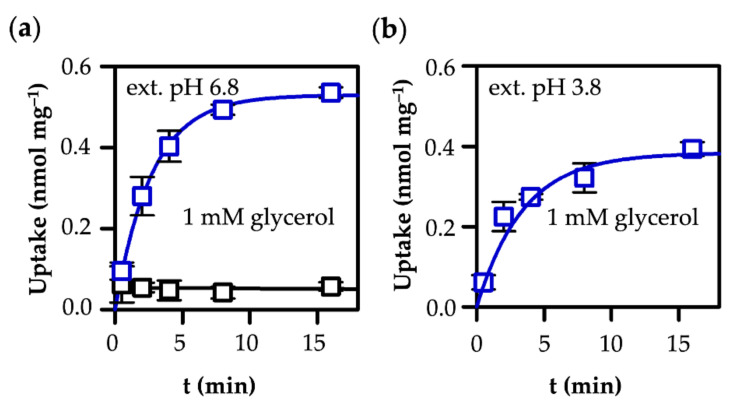
Uptake of glycerol into human AQP9-expressing yeast at pH 6.8 (**a**) and 3.8 (**b**). The level of the background of non-expressing cells (black) is indicated in (**a**) and was subtracted from the uptake curves.

**Figure 3 life-12-00120-f003:**
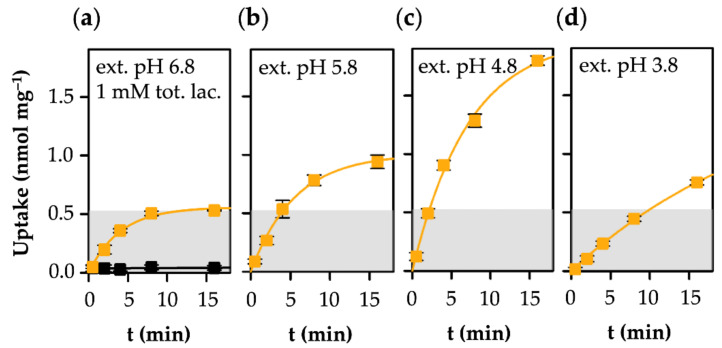
Uptake of l-lactate into human MCT1-expressing yeast at pH 6.8 (**a**), 5.8 (**b**), 4.8 (**c**), and 3.8 (**d**). The external l-lactate/l-lactic acid concentration was 1 mM. The resulting free l-lactate species concentrations are 0.999, 0.99, 0.9, and 0.5 mM, respectively. The top of the gray shaded area indicates the 1 mM cytosolic substrate concentration level. The level of the background of non-expressing cells (black) is indicated in (**a**) and was subtracted from the uptake curves.

**Figure 4 life-12-00120-f004:**
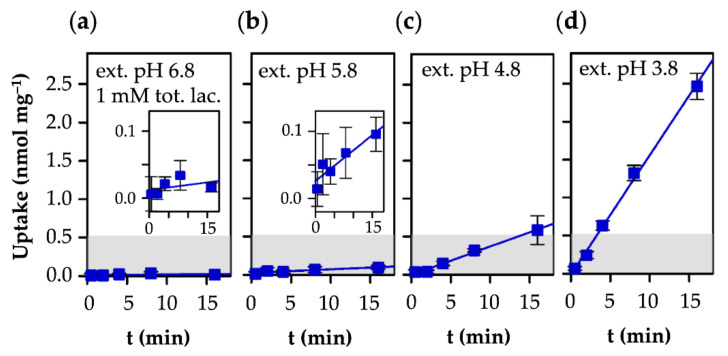
Uptake of l-lactate into human AQP9-expressing yeast at pH 6.8 (**a**), 5.8 (**b**), 4.8 (**c**), and 3.8 (**d**). The external l-lactate/l-lactic acid concentration was 1 mM. The resulting free l-lactic acid species concentrations are 0.001, 0.01, 0.1, and 0.5 mM, respectively. The top of the gray shaded area indicates the 1 mM cytosolic substrate concentration level.

**Figure 5 life-12-00120-f005:**
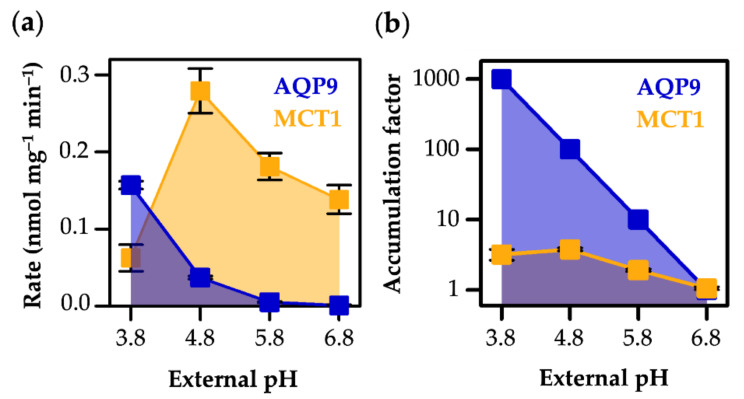
pH gradient-dependent uptake rate of l-lactate/l-lactic acid (**a**) via AQP9 (blue) and MCT1 (orange) and accumulation (**b**: note the logarithmic scale of the ordinate). The accumulation factors for MCT1 transport were derived from the plateaus of the uptake assay, those for AQP9 were calculated based on the l-lactate/l-lactic acid protonation equilibrium.

**Figure 6 life-12-00120-f006:**
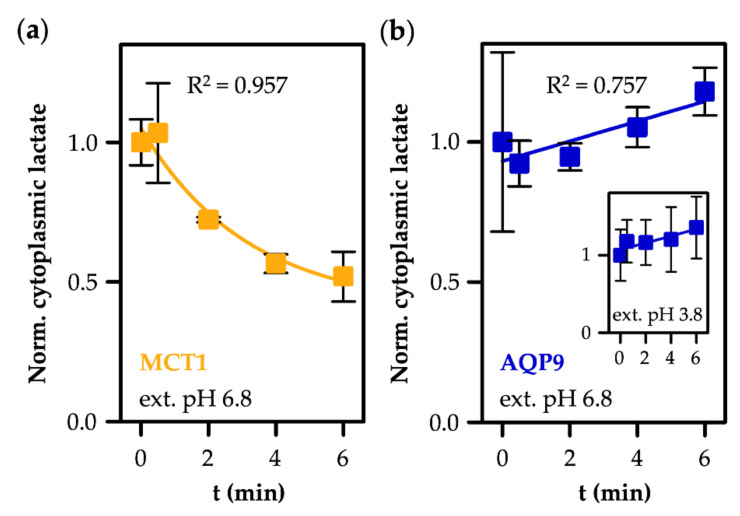
Efflux of l-lactate/l-lactic acid via MCT1 (**a**) and AQP9 (**b**). Cells were loaded in 1 mM ^14^C-l-lactate/l-lactic acid buffer (pH 3.8), collected and resuspended in substrate-free buffer of pH 6.8 (the inset in (**b**) depicts absence of efflux via AQP9 also at an external buffer pH of 3.8). Shown is the remaining cytosolic l-lactate/l-lactic acid relative to the initial concentration after loading.

**Figure 7 life-12-00120-f007:**
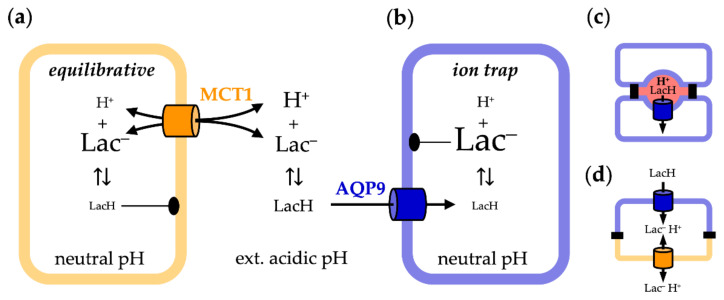
Cellular uptake of l-lactate/H^+^ via MCT1 (**a**, Lac^−^/H^+^) and l-lactic acid via AQP9 (**b**, LacH). The resulting distribution of the substrate species in a transmembrane pH gradient is symbolized by the font size. (**c**,**d**) Scenarios in which AQP9-mediated l-lactic acid permeability may be of relevance; see text for details.

## Data Availability

The data presented in this study are fully contained in the paper.

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
