# Peer review of "Lactic Acid Permeability of Aquaporin-9 Enables Cytoplasmic Lactate Accumulation via an Ion Trap"

_life, 2022, doi:10.3390/life12010120_

Round 1
Reviewer 1 Report
Human aquaporin-9 (AQP9) conducts several small uncharged metabolites, such as glycerol and lactic acid. To explore the physiological significance of upregulated AQP9 expression in certain brain tumors with respect to increased lactic acid uptake, Geistlinger et al. explore the contribution of human AQP9 and monocarboxylate transporter 1 (MCT1) to L-lactate/L-lactic acid transport at different external pH conditions by expressing these proteins in yeast. The results demonstrated that while transport via MCT is equilibrative with respect to the extracellular L-lactate and proton concentrations, the L-lactic acid permeability of AQP9 enables an ion trap leading to a large accumulation of intracellular substrate over the extracellular concentration at steep transmembrane pH conditions. Thus, the lactic permeability of AQP9 may be only physiologically relevant when the transmembrane proton gradient is steep, and AQP9 acts as the sole uptake path at least on one side of a polarized cell. Overall, the experiments are well designed, the results are solid and conclusions are convincing.
Minor suggestions:
- Line 72: “expression constructs of human AQP9 (NCBI Gene ID 366) and human MCT1 (GenBank NM_001166496) in the modified pDR196 vector, containing the URA3 gene, were generated previously [15,23].” Please indicate whether AQP9 and MCT1 proteins have been verified to be expressed in yeast in the previous studies. If not, the verified expression on protein levels is needed.
- As glycerol and lactic acid may also permeate freely through the cellular membrane, although the authors have subtracted the background information in the control strain for the data in Figs. 2, 3, 4 and 6, the data of the controls strain are better to be provided in the figures.
Author Response
Minor suggestions:
- Line 72: “expression constructs of human AQP9 (NCBI Gene ID 366) and human MCT1 (GenBank NM_001166496) in the modified pDR196 vector, containing the URA3 gene, were generated previously [15,23].” Please indicate whether AQP9 and MCT1 proteins have been verified to be expressed in yeast in the previous studies. If not, the verified expression on protein levels is needed.
>> We have shown expression of the AQP9/MCT1 constructs in three earlier publications and therefore added a sentence referring to the Western blots in the previous papers.
- As glycerol and lactic acid may also permeate freely through the cellular membrane, although the authors have subtracted the background information in the control strain for the data in Figs. 2, 3, 4 and 6, the data of the controls strain are better to be provided in the figures.
>> We added background glycerol and lactic acid diffusion data to Figs. 2 and 3 to visualise their minimal contribution to the AQP9/MCT1-facilitated uptake. We further explicitly the background level as numbers in the Methods section.
Reviewer 2 Report
This manuscript describes the relevance of AQP9 to lactic acid transmembrane permeability and accumulation. The authors showed that AQP mediated uptake of lactic acid depends on a molecular form of lactic acid based on external pH conditions, and AQP9 course cytoplasmic lactic acid/lactate accumulation linearly at pH around the pKa of lactic acid. Although it is difficult to understand the relevance of AQP9 to lactic acid accumulation in brain tumors by the authors works these results may help to understand to mechanism AQP9, so this manuscript is recommended for publishing after minor revision.
Minor revisions.
Fig 6b: The addition of an explanation for why the exposure of the cells to a substrate-free buffer indicated an increase of lactic acid level is helpful for readers to understand.
Fig 7c and d: I wonder these situations exist within the human body. Could you cite some paper to support Fig 7c and d?
Author Response
Fig 6b: The addition of an explanation for why the exposure of the cells to a substrate-free buffer indicated an increase of lactic acid level is helpful for readers to understand.
>> In fact there is no increase over time if one considers the error margins of the assay. In order to indicate this better, we added the correlation coefficients (R^2) to the figure. While the MCT1 efflux curve closely followed a single exponential fit (R^2 = 0.957), the AQP9 assays exhibited larger variation leading to a lower R^2 of 0.757; the resulting error margin of 1 +/– 0.3 has been added to the text.
Fig 7c and d: I wonder these situations exist within the human body. Could you cite some paper to support Fig 7c and d?
>> So far, the described situations are speculative. We changed the wording to make this point more clear. There are only few studies on the AQP9 expression and exact subcellular localisation in the brain and in tumors. Our biophysical data are meant to point out what to look for if one plans a respective study.
Reviewer 3 Report
In this manuscript the Authors explored contribution of AQP9 to lactate transport in mutant yeast strain lacking functional monocarboxylate transporters. The main experimental findings of the work are: (1) AQP9 facilitates accumulation and export of L-lactic acid in its neutral form [not novel]. (2) Under select conditions, maximal capacity of AQP9 for L-lactate accumulation is THOUGHT to be magnitude+ higher than the monocarboxylate transporter MCT1. The second idea/finding is an interesting notion that is not fully supported by experimental data.
The major idea of the work is summarized in theoretical Fig. 5b, which suggests that due to charge trapping, AQP9 provides low-velocity but high-capacity system for L-lactate uptake. It is further partially tested in experiments presented in Fig. 6, indicating that AQP9-expressing cells trap lactate while MCT1-expressing cells equilibrate it. While this work is methodologically sound and conceptually interesting, I have several concerns. These are related to the caveats of interpretation, which are listed below.
Specific comments:
[1] At near physiological pH values (in this paper pH=6.8), AQP9-expressing cells (Fig. 3a) accumulate tiny fraction of [14C]lactate as compared to MCT1-expressing cells (Fig. 4a). The Authors perform “thought experiment” extrapolating the linear [14C]lactate uptake shown in Fig. 4 and the apparent lack of [14C]lactate efflux in Fig. 6b to make major theoretical statements about lactate trapping and high capacity of lactate accumulation via AQP9. This is confounded by low accuracy of transport measurements in the AQP9-expressing cells. I ask the Authors to directly test their conclusion by measuring L-[14C]lactate accumulation in yeast over longer periods of time. With “linear” uptake, yeast are expected to accumulate 4x [14C]lactate quantities after 1 hr incubation and 8x levels after 2 hr (as compared to the presented experiments). Please test it experimentally.
[2] The Authors make interesting assumptions about contribution of AQP9 to lactate redistribution in brain tumor environment, such as in glioblastomas (GBM). Based on the Authors results and in line with their discussion, the presence of even low levels of MCTs in tumor cells (EXPECTED), makes the AQP9-driven lactate accumulation a moot point. The Authors then propose the polarity of MCT and AQP9 expression, which in opinion of this reviewer is neither grounded in any prior observations nor physiologically meaningful.
[3] I think that interpretation of potential AQP9 functions in the context of L-lactate transport is too myopic. This transporter protein has been studies in the context of several major physiological substrates (glycerol, etc.). This is mentioned in Introduction but not properly discussed. At the very least, the Authors need to be open to a possibility that AQP9 expression serves many lactate-unrelated roles, in tumor environment and elsewhere.
[4] I have a moderate concern about scientific rigor of experiments. The Authors performed efflux experiments (Fig. 6) in biological duplicates. I presume that the data presented in this figure represent variability of technical replicates in one experiment. This is most likely insufficient for making quantitative or qualitative conclusions.
Author Response
[1] At near physiological pH values (in this paper pH=6.8), AQP9-expressing cells (Fig. 3a) accumulate tiny fraction of [14C]lactate as compared to MCT1-expressing cells (Fig. 4a). The Authors perform “thought experiment” extrapolating the linear [14C]lactate uptake shown in Fig. 4 and the apparent lack of [14C]lactate efflux in Fig. 6b to make major theoretical statements about lactate trapping and high capacity of lactate accumulation via AQP9. This is confounded by low accuracy of transport measurements in the AQP9-expressing cells. I ask the Authors to directly test their conclusion by measuring L-[14C]lactate accumulation in yeast over longer periods of time. With “linear” uptake, yeast are expected to accumulate 4x [14C]lactate quantities after 1 hr incubation and 8x levels after 2 hr (as compared to the presented experiments). Please test it experimentally.
>> We would have liked to assay the uptake for longer periods if this was possible. The expression systems apart from yeast, such as human cells or Xenopus oocytes allow for very short assay times in the seconds time scale due to rapid energy metabolism. With the yeast we managed to extend the time frame to 18 min, which was typically sufficient for MCT1 measurements and safe extrapolations. In the case of AQP9, the assay time was long enough to show linearity in this range and to show the higher uptake capacity (compare the scale of the ordinates in Fig. 3 and 4). The fitness of the yeast cells under the assay conditions precluded longer measurements. We added text to explain the advantages and restrictions to the Results section plus references to earlier work.
[2] The Authors make interesting assumptions about contribution of AQP9 to lactate redistribution in brain tumor environment, such as in glioblastomas (GBM). Based on the Authors results and in line with their discussion, the presence of even low levels of MCTs in tumor cells (EXPECTED), makes the AQP9-driven lactate accumulation a moot point. The Authors then propose the polarity of MCT and AQP9 expression, which in opinion of this reviewer is neither grounded in any prior observations nor physiologically meaningful.
>> This point was also raised by reviewer 1: So far, the described situations are speculative. We changed the wording to make this point more clear. There are only few studies on the AQP9 expression and exact subcellular localisation in the brain and in tumors. Our biophysical data are meant to point out what to look for if one plans a respective study.
[3] I think that interpretation of potential AQP9 functions in the context of L-lactate transport is too myopic. This transporter protein has been studies in the context of several major physiological substrates (glycerol, etc.). This is mentioned in Introduction but not properly discussed. At the very least, the Authors need to be open to a possibility that AQP9 expression serves many lactate-unrelated roles, in tumor environment and elsewhere.
>> We appreciate this comment and agree that the respective part of the discussion was too much focussed on lactic acid. We added text and a reference to a review article to bring in other aspects of putative AQP functionality in tumors, namely glycerol clearance and metastasis.
[4] I have a moderate concern about scientific rigor of experiments. The Authors performed efflux experiments (Fig. 6) in biological duplicates. I presume that the data presented in this figure represent variability of technical replicates in one experiment. This is most likely insufficient for making quantitative or qualitative conclusions.
>> Each experiment was done with two independently grown biological samples, each at two different pH conditions and two technical replicates each. That means for each time point 2^3 = 8 measurements were available. Further, contrary to MCT1-expressing cells, with AQP9 all time points yielded the same value within the error range providing another indication of coherence throughout the experiment.
Round 2
Reviewer 3 Report
I am satisfied with the Authors' responses and revisions.